

# Unsupervised learning analysis on the proteomes of Zika virus

Edgar E. Lara-Ramírez[1], Gildardo Rivera[1], Amanda Alejandra Oliva-Hernández[2], Virgilio Bocanegra-Garcia[3], Jesús Adrián López[4] and Xianwu Guo[5]

[1] Laboratorio de Biotecnología Farmacéutica, Centro de Biotecnología Genómica, Instituto Politécnico Nacional, Reynosa, Tamaulipas, México

[2] Laboratorio de Biotecnología Experimental, Centro de Biotecnología Genómica, Instituto Politécnico Nacional, Reynosa, Tamaulipas, México

[3] Laboratorio de Interacción Ambiente Microorganismo, Centro de Biotecnología Genómica, Instituto Politécnico Nacional, Reynosa, Tamaulipas, México

[4] Laboratorio de microRNAs y Cáncer, Unidad Académica de Ciencias Biológicas, Universidad Autónoma de Zacatecas, Zacatecas, Zacatecas, México

[5] Laboratorio de Biotecnología Genómica, Centro de Biotecnología Genómica, Instituto Politécnico Nacional, Reynosa, Tamaulipas, México

Corresponding authors
Edgar E. Lara-Ramírez,
doc_lara_ram@hotmail.com
Xianwu Guo, xguo@ipn.mx

## ABSTRACT

**Background**. The Zika virus (ZIKV), which is transmitted by mosquito vectors to nonhuman primates and humans, causes devastating outbreaks in the poorest tropical regions of the world. Molecular epidemiology, supported by clustering phylogenetic gold standard studies using sequence data, has provided valuable information for tracking and controlling the spread of ZIKV. Unsupervised learning (UL), a form of machine learning algorithm, can be applied on the datasets without the need of known information for training.

**Methods**. In this work, unsupervised Random Forest (URF), followed by the application of dimensional reduction algorithms such as principal component analysis (PCA), Uniform Manifold Approximation and Projection (UMAP), t-distributed stochastic neighbor embedding (t-SNE), and autoencoders were used to uncover hidden patterns from polymorphic amino acid sites extracted on the proteome ZIKV multi-alignments, without the need of an underlying evolutionary model.

**Results**. The four UL algorithms revealed specific host and geographical clustering patterns for ZIKV. Among the four dimensionality reduction (DR) algorithms, the performance was better for UMAP. The four algorithms allowed the identification of imported viruses for specific geographical clusters. The UL dimension coordinates showed a significant correlation with phylogenetic tree branch lengths and significant phylogenetic dependence in Abouheif's Cmean and Pagel's Lambda tests (p value < 0.01) that showed comparable performance with the phylogenetic method. This analytical strategy was generalizable to an external large dengue type 2 dataset.

**Conclusion**. These UL algorithms could be practical evolutionary analytical techniques to track the dispersal of viral pathogens.

# INTRODUCTION

The Zika virus (ZIKV) illness is a neglected tropical disease that causes devastating outbreaks in the poorest populations of the world (*Sarkar & Gardner, 2016*). In humans, ZIKV infection causes mild or asymptomatic disease; however, the infection has been associated with miscarriage and congenital microcephaly in the newborns who acquired the virus in any trimester during the pregnancy of women and associated with the neurological Guillain Barré syndrome in adults (*Rawal, Yadav & Kumar, 2016*). ZIKV is transmitted to nonhuman primate (NHP) and human hosts by different mosquito vectors (*Terzian et al., 2018*). In NHP, the virus follows a sylvatic cycle where the primary vectors are arboreal *Aedes* spp.; meanwhile, in humans, the virus follows the urban cycle where the primary vectors are peridomestic/domestic *Aedes* spp (*Vasilakis & Weaver, 2017*).

ZIKV is in the Flavivirus genus, together with the type 1–4 dengue virus and yellow fever virus, which are also transmitted by the vector *Aedes aegypti* (*Souza-Neto, Powell & Bonizzoni, 2019*). The genome is composed of a single-stranded positive RNA of 10,716 nucleotides that encodes a polyprotein of 3,000 amino acids in length. The proteome comprises the capsid (C), pro-envelope (prM), and envelope (E) structural proteins and seven non-structural proteins NS 1, 2a, 2b, 3, 4a, 4b, 5. Phylogenetic clustering studies using these sequence data allowed to trace the evolution of the pathogen and provided valuable information to monitor the dispersal of the pathogen (*Hung & Huang, 2021*).

**Motivation:** Clustering approaches exist, other than phylogenetic methods, such as the unsupervised learning (UL) algorithms, a subfield of the machine learning study area. The UL approach facilitates to discover meaningful cluster structures on the data without the need of training the algorithm with known information. For example, unsupervised random forest (URF) has been applied successfully in an earlier study to identify clusters related to human and mouse species using data pertaining to random monoallelic and long interspersed nuclear element imprinted genomic sequences (*Allen et al., 2003*). This approach was also used to distinguish tumor clusters related to immunohistochemistry expression markers (*Shi & Horvath, 2006*). Recently, the URF approach was also used to identify patients with coronary artery disease using clinical data for the learning algorithm (*Zanfardino et al., 2024*). Although URF has proved its successful application in different research areas, this approach is not explored in the analysis of genomic and proteomic sequence data. The main assumption of the URF approach is its capability of distinguishing original data from its derived synthetic data, if the results are meaningful, the subsequent analysis based the production of a proximity matrix that can be used as input to other dimensionality reduction (DR) should recover hidden structures on the real data (*Afanador et al., 2016*). Regarding the DR algorithm, there are several options, of which, principal component analysis (PCA) is the oldest and most popular DR algorithm, assuming linear relations on the data. On the contrary, the new DR alternatives such as Uniform Manifold Approximation and Projection (UMAP), t-distributed stochastic neighbor embedding (t-SNE), and autoencoders based on deep learning (*Lin, Mukherjee & Kannan, 2020*) assumed a nonlinear structure on the data. The final aim of all these DR algorithms is to reduce the original data to a manageable spatial arrangement of clusters that can be

visually analyzed in two dimensions or axes. In this work, our objective was to apply these distinct UL clustering approaches to analyze the evolution of ZIKV at proteome level and to compare the results with the gold standard clustering phylogenetic methods.

**Reproducibility:** The R code implementing the unsupervised analysis is available in the supplementary materials.

## MATERIALS & METHODS

### Data preprocessing

The whole flow chart is shown in the Fig. 1. It started by retrieving a total of 391 nonredundant ZIKV proteomes along with their metadata (available as Supplementary Materials) for hosts (human, mosquito, and primate) and geographic regions from the virus variation database (https://www.ncbi.nlm.nih.gov/genomes/VirusVariation/Database/nph-select.cgi?cmd=database&, accessed on 08-09-2023) (*Brister et al., 2013*). After eliminating proteomes with unambiguous amino acids, the final data set was composed of 293 proteomes. This sequence data was employed to construct multi alignments with the MAFFT (Multiple Alignment using Fast Fourier Transform) software (*Katoh, Rozewicki & Yamada, 2019*) using default options (available as Supplementary Materials). The proteome phylogenetic tree was constructed from the multi-alignment with IQ-TREE v. 2.3.6 software, which select the most appropriate amino acid evolution mathematical model (*Minh et al., 2020*) (http://www.iqtree.org/). The resulting newick format tree file was plotted and colored based on the geographic metadata with the iTOL web platform (https://itol.embl.de/) (*Letunic & Bork, 2021*) (available Supplementary Materials).

### Description of used models

URF: The ML algorithms were implemented in house-scripts running on R 4.2.2 version (*R Development Core Team, 2016*) as follows: First the amino acid polymorphic sites were extracted from the multi-alignments with the function "alignment2genind" from the adegenet R package (*Jombart, Balloux & Dray, 2010*), from which 422 polymorphic sites were determined. Based on these data, a synthetic data set of the same size was produced by permutating the rows of each column and was then combined with the original data. This information was used to perform a Random Forest (RF) analysis with the R package randomForest using the function "randomForest" (*Breiman, 2001*; *Liaw & Wiener, 2002*). The hyperparameters were optimized using a five-fold cross-validation with three-repeat procedure with the caret R package (*Kuhn, 2008*). The "mtry" and "ntree" parameters were settled using grid search, that include the square root (sqrt) of the predictors and 500, 1,000, 1,500, and 2,000 trees. The RF out of bag (OOB) error was used as partial validation of goodness of fit for the URF. The final best hyperparameters were then used to perform the URF on the original data by including the parameter proximity = TRUE, which allowed to obtain the similarity (or proximity) matrix. The produced matrix was converted to a dissimilarity matrix by subtracting 1 for each data to compress all the information as previously described (*Liaw & Wiener, 2002*; *Afanador et al., 2016*). Then the matrix was converted to a Euclidean distance matrix to confirm the structure on the data *via* heatmap plotting and the Hopkins statistic. Finally, the dissimilarity matrix based on real data

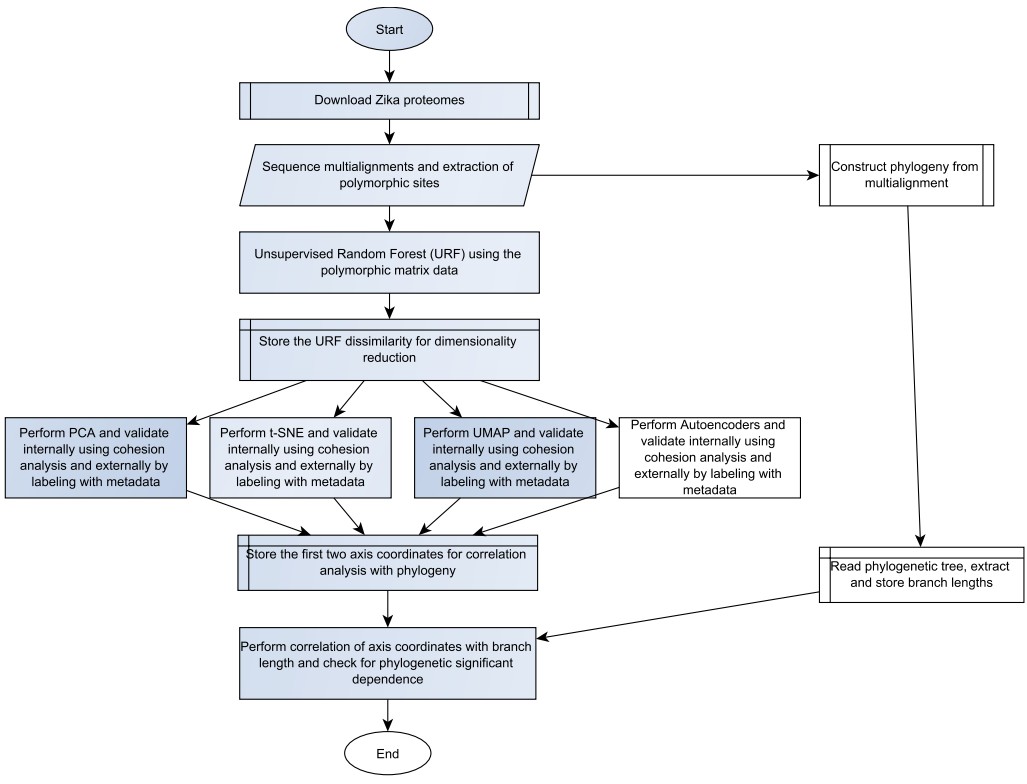

**Figure 1  The analytical workflow of this study.**

was further used as input to the PCA, t-SNE, UMAP, and autoencoders dimensionality reduction algorithms. The pseudocode that includes the URF and the DR procedures is shown below.

1. Combined_Data=real_data + synthetic_data,
2. Unsupervised RandomForest(data=combined_data, tree=num_trees, mtry=min_samples_split, proximity= proximity_matrix):
3. forest = list()
4. proximity = matrix(len(data), len(data))
5. For i in tree do:
6. bootstrap_sample = bootstrap(data)
7. tree = randomForest(bootstrap_sample, mtry)
8. forest.aggregate (tree)
9. For each pair of samples (x_i, x_j) in data do:
   If x_i and x_j end up in the same leaf node in the tree then:
   proximity_matrix[i][j] += 1
   End If
   End For
   return (forest, proximity_matrix)
   return (proximity_matrix/ntree)

11. evaluate error rate on OOB
12. UnsupervisedRandomForest(data=real_data, tree=num_trees, mtry=min_samples_split, proximity=TRUE):
13. distance_matrix = 1 - proximity_matrix
14. cluster with PCA(distance_matrix)
15. cluster with t-SNE(distance_matrix)
16. cluster with UMAP(distance_matrix)
17. cluster with Autoencoders(distance_matrix)

PCA: The PCA was applied using the function "prcomp" and the option "scale=FALSE" with the R base package. PCA uses scaled data to calculate the covariance matrix to identify correlated variables. Then this information is used to derive new uncorrelated variables called the principal components that retain the variance of the original data into few axes' coordinates.

t-SNE: The t-SNE algorithm was applied with "Rtsne" function of the Rtsne package by settling the perplexity parameter at 50 with 1000 step iterations to identify the similar and dissimilar data points. t-SNE reduces the high dimensional data using a symmetrized cost function and Student-t distribution to calculate the similarity in the low-dimensional space of two data points randomizing the initialization (*Van der Maaten, 2014*; *Krijthe, 2015*). To balance local and global features in the data, t-SNE uses the perplexity parameter that varies in the range of 5 to 50 to guess the closest neighbors for each data point.

UMAP: UMAP was implemented with the "umap" function from the umap R package that considers 200 epochs and 15 neighbors as cut-off to compute the similarity. The UMAP algorithm is similar to the t-SNE and uses the t-distribution to calculate the similarity in the data except that UMAP initializes in the same data points through spectral embeddings that allows to move a single or group of data points at once to calculate its similarity, making it faster than t-SNE (*McInnes et al., 2018*).

AutoEncoders: Autoencoders neural network was constructed using the Keras package for R (*Chollet, 2015*). The neural architecture was designed containing three layers with 12 neurons in the first encoder layer, six in the second bottleneck layer and 12 in the third decoder layer using the activation hyperbolic tangent function settled with the parameter "tanh". Using these parameters the AE shrinks the data into the bottleneck with 2,000 training iteration epochs.

## Validation

External validation was carried visually by labeling the recovered clusters with the metadata information through two-dimensional plots resolved by each DR; thus, if the clusters formed according to its geography and host isolation, they were considered true clusters. In addition, the internal validation was performed through silhouette plots using k-means and the Euclidean distance in the elbow analysis to identify the number of suitable groups produced by each DR. The silhouette coefficient was considered the mean distance between samples in specific groups and other clusters. Thus, the larger the coefficient, the more consistency is in the cluster distribution. Finally, the whole unsupervised analysis strategy was applied to a larger dataset retrieved from virus variation

(https://www.ncbi.nlm.nih.gov/genomes/VirusVariation/Database/nph-select.cgi?cmd=database&, accessed on 26-07-2024) comprising 1,343 whole dengue type 2 (DENV2) proteomes isolated from human hosts (available as supplementary materials). As these DR methods could produce slightly different results in each training test, six test runs were performed and analyzed to confirm the reproducibility of the cluster patterns.

### Phylogenetic correlation and dependence test analysis

The R packages ape and adephylo (*Jombart, Balloux & Dray, 2010*) were used to read and merge the axes cluster coordinates with the phylogeny using the function "phylo4d". Then the phylogenetic correlations and statistical test were calculated with the package phylosignal (*Keck et al., 2016*). This package contains several functions to calculate correlations and statistical test of phylogenetic dependence of continuous trait values with the phylogenetic branches. The phylogenetic correlograms were produced with the function "phyloCorrelogram". The Abouheif's Cmean and Pagel's Lambda tests were calculated with the functions "phyloSignal" specifying the methods "Cmean" and "Lambda" for each test, respectively. A *p* value <0.01 was deemed as significant in the phylogenetic statistical test under the null hypothesis of lack of phylogenetic dependence.

## RESULTS

### Unsupervised random forest retrieves cluster tendency

Using the best hyperparameters (mtry=10, ntree=1,500, Fig. S1) the RF indicates an OOB of 1.19%, confirming that it can distinguish the real data from its synthetic version. Moreover, the Hopkins statistic value was of 0.91, indicating the existence of clustering tendency in the real data. The visual exploration confirmed the cluster tendency which was according to the geographic origin (Fig. 2). In the heatmap six well-defined blocks are formed, using the Euclidean distance.

### UMAP showed the best performance for cluster separation

To prove the clustering performance for PCA, t-SNE, UMAP and Autoencoders, the k-means clustering using the elbow curve analysis based on silhouette coefficient was applied. The analysis of the elbow curve showed similar separation distances (average silhouette coefficient >0.6, Figs. 3A, 3D, 3E and 3G) among the four tested DRs, but different number of clusters of k (range 3 to 8). Among them, PCA and UMAP were the best showing the ability to identify more clusters than the other algorithms (Figs. 3B, 3D, 3F and 3H), but UMAP showed better average silhouette coefficient and cluster separation than PCA (Figs. 3E and 3F).

### Geography is the main factor for cluster separation

After labeling the clusters, the four algorithms showed that ZIKV isolates were grouped according to their host (Fig. 4). For example, those from arboreal mosquitoes clustered with NHP ZIKV isolates and the *Aedes aegypti* isolates clustered mainly with human isolates. In the cluster analysis, the African ZIKV group was clearly separated from the other geographical regions, as it consisted of NHP and arboreal mosquitoes and the

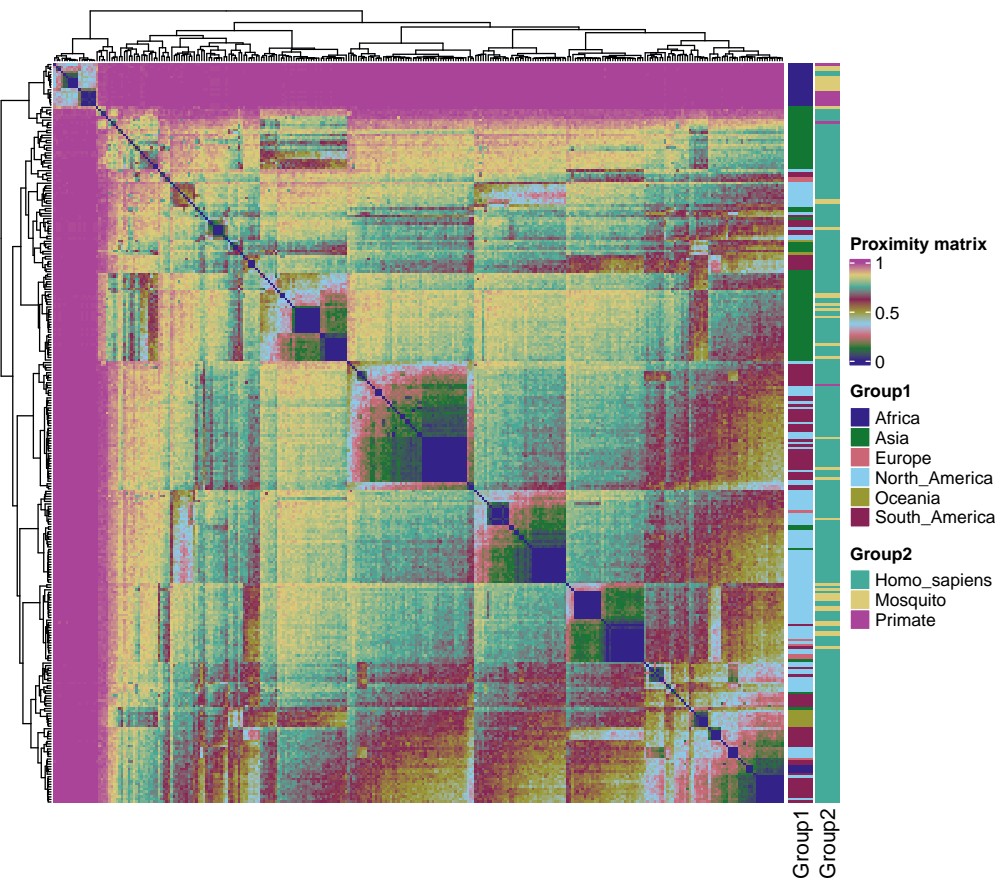

**Figure 2** **Heatmap based on the proximity matrix obtained in the amino acid polymorphic sites produce by the URF.** The right bar colors indicate the groups by geographic regions (group 1) and host origins (group 2). The muted nine-style colorblind-friendly (https://github.com/JLSteenwyk/ggpubfigs) in the blocks indicates the similarity within the groups.

human isolate (virus variation ID AMR68906), supporting the view of limited adaptation of sylvatic strains compared to endemic urban strains (*Beaver et al., 2018*). Interestingly, a clear geographical dispersion pattern was observed for some Asian ZIKV that tend to migrate into the European, Oceanic, North American, and South American regions. Furthermore, these Asian isolates tend to mix tightly with some American isolates. The t-SNE and UMAP and showed the clearest separation within regional clusters, in which the North American cluster harboring few Asian genomes was of note (Figs. 4D and 5F). In this cluster, the DR analyzes identified three Asian strains (sequence IDs ASU55505, ATI21641, and ARI68105), which were previously identified as imported American strains in Hainan China and Singapore (*Tan et al., 2018*).

## The analytical strategy was generalizable to an external proteome dataset

To confirm whether the analytical strategy for unsupervised analysis takes a broad view to other real datasets, we applied the whole analytical strategy to a data matrix of polymorphic

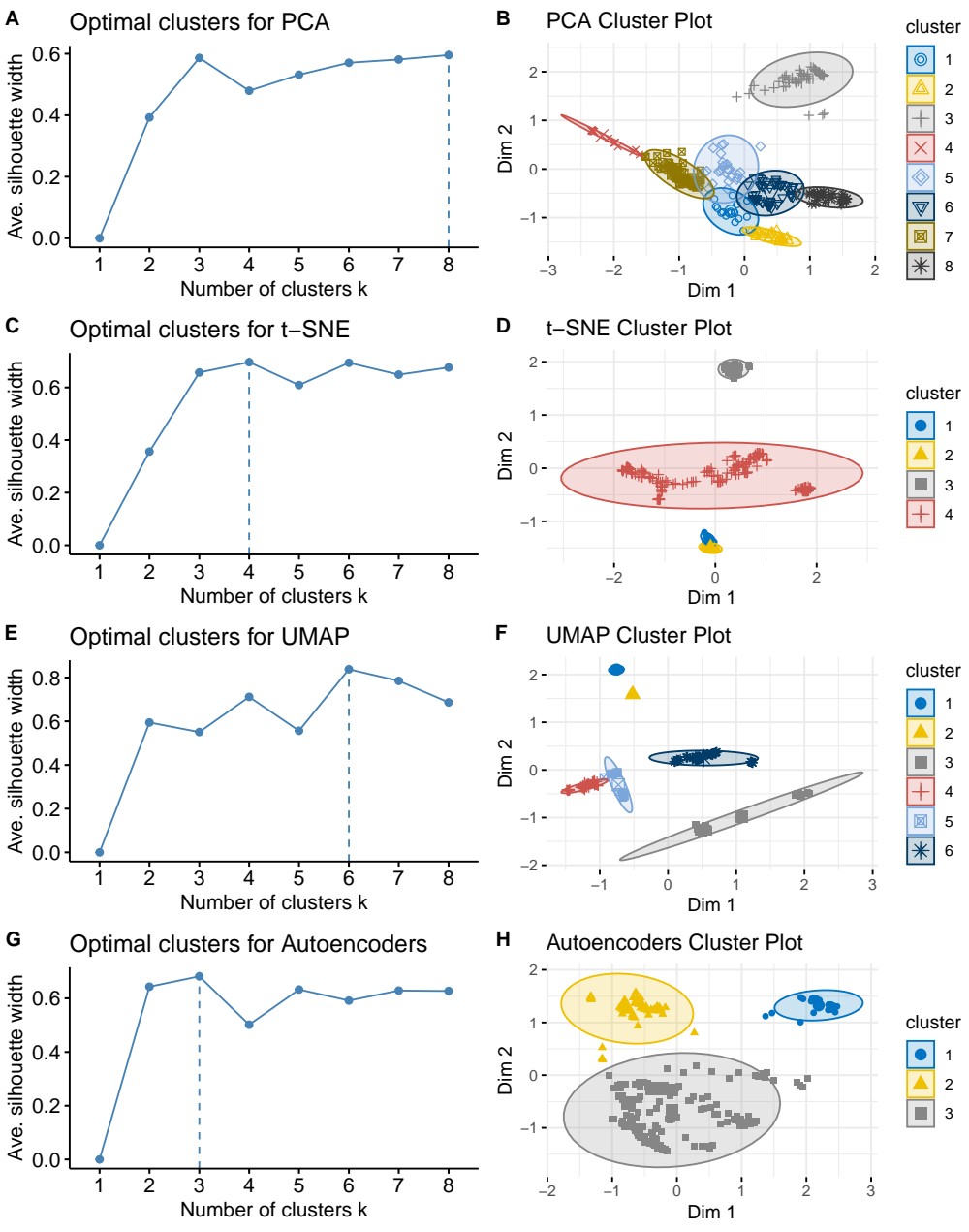

**Figure 3** **Internal validation of clustering performance for each of the four DR algorithms.** This figure showed the elbow curves (A, C, E, G) and cluster (B, D, F, H) plots based on k means clustering using Euclidean distance.

sites extracted from a multialignment of 1,343 DENV2 proteomes. We found that the dissimilarity matrix produced by the URF recovers the cluster tendency (Fig. S2) with a Hopkins statistic of 0.93. The UMAP was also the best DR algorithm compared to the others (Fig. S3, Fig. 5C). External validation by labeling with metadata also showed that geography is the main factor shaping the DENV2 dispersion (Figs. 5A, 5B, 5C and 5D), but

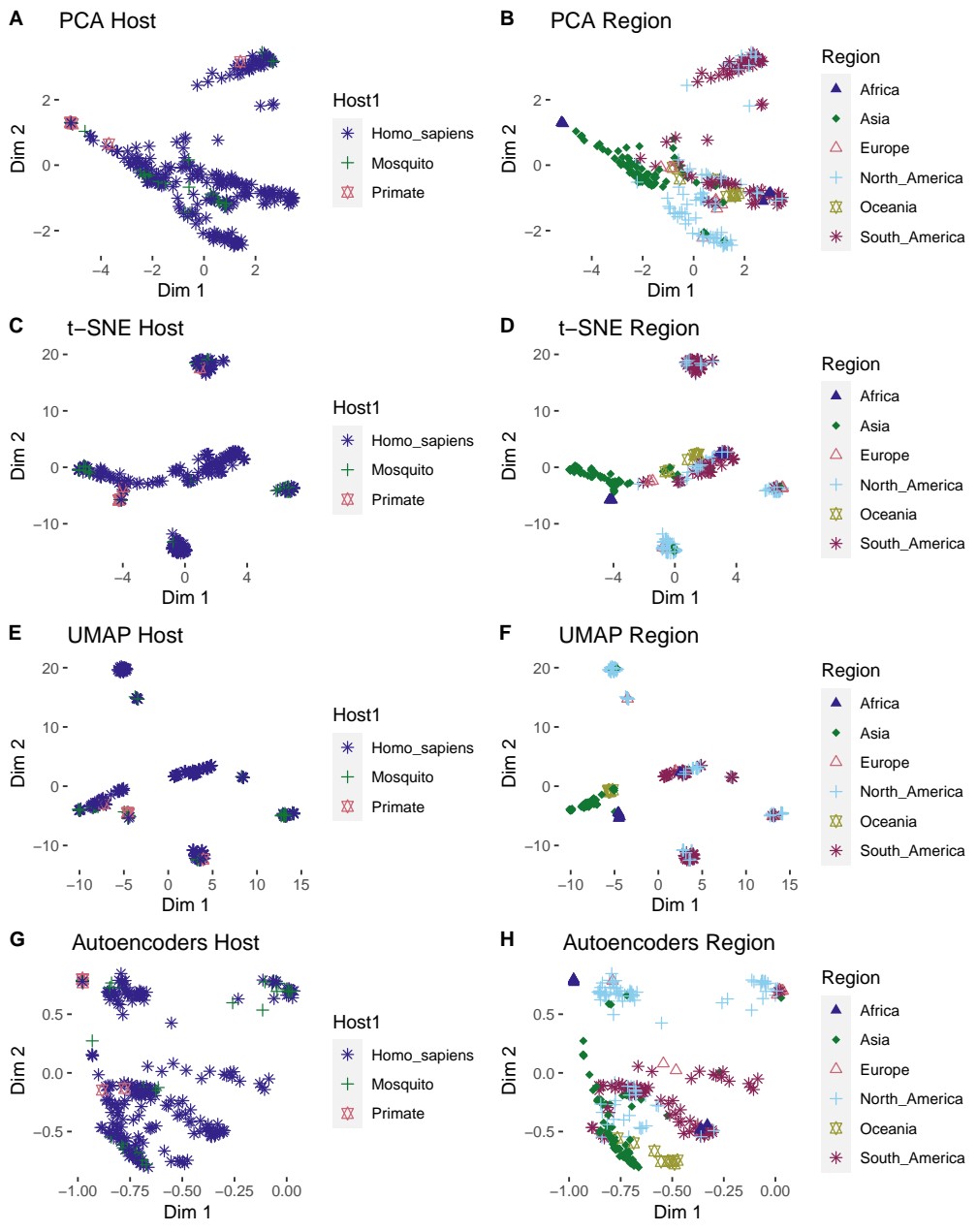

**Figure 4** **External validation by ZIKV metadata labeling for the four DR algorithms.** On the panels the clusters produced by scattering the first two dimensions (dim1 and dim2) represent the ZIKV proteomes as colored points and shapes by its metadata information as host (A, C, E, G) or geographic isolation regions (B, D, F, H).

was particularly evident in the UMAP (Fig. 5C). This result confirmed that this analytical strategy could be generalized to analyze the dispersal of viral pathogens.

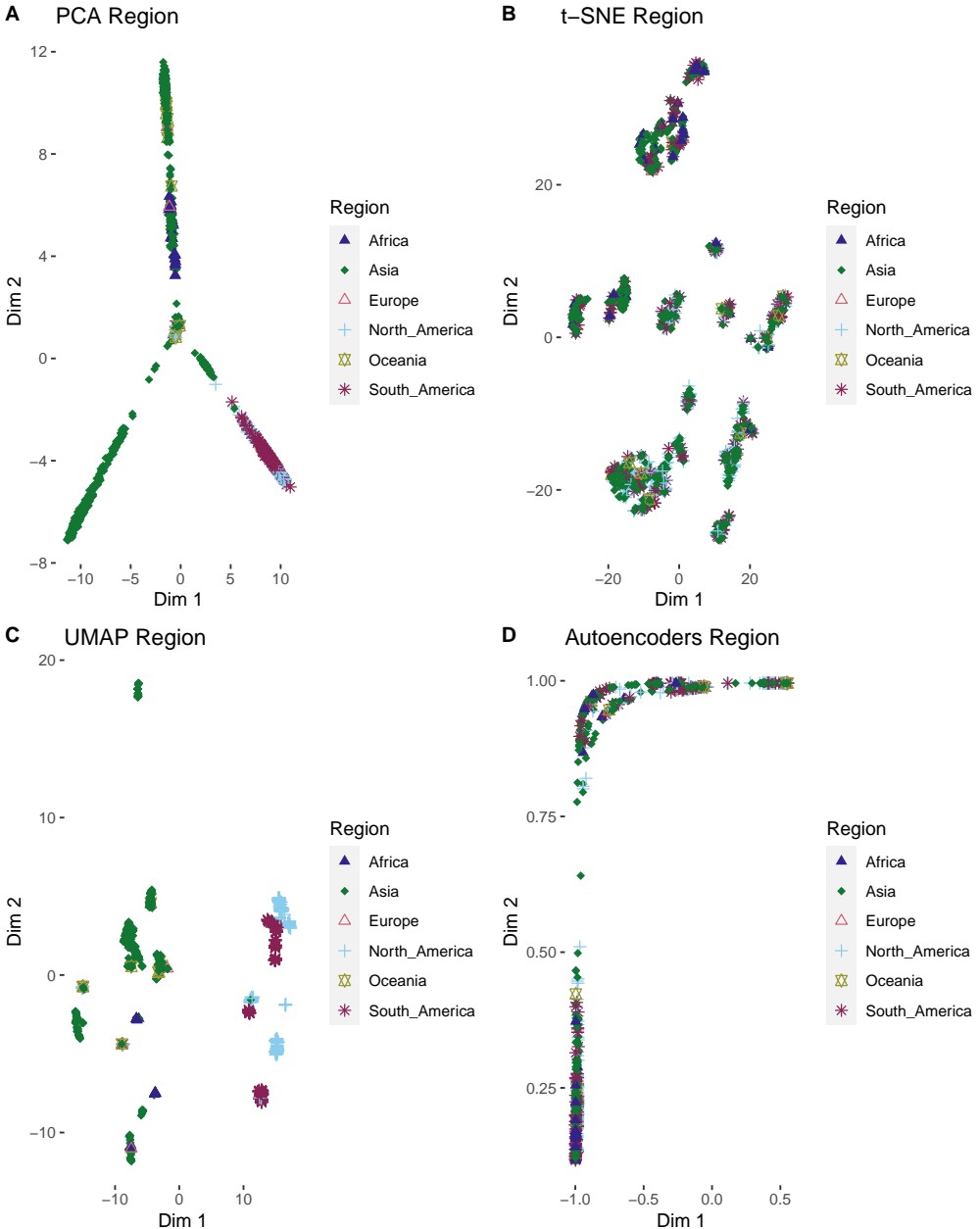

**Figure 5** **DENV2 external validation for four DR algorithms.** On the panels the clusters produced by scattering the first two dimensions (dim1 and dim2) represent the DENV proteomes as colored points and shapes by its geographic metadata (A, B, C, D).

## Phylogeny correlates with cluster coordinates

As expected, the phylogeny showed similar clustering tendencies; for instance, the African ZIKV were clearly separated from the other groups with a larger branching length. The same Asian ZIKV were mixed with European, Oceanic, North American, and South American regions with smaller branching length (Fig. 6).

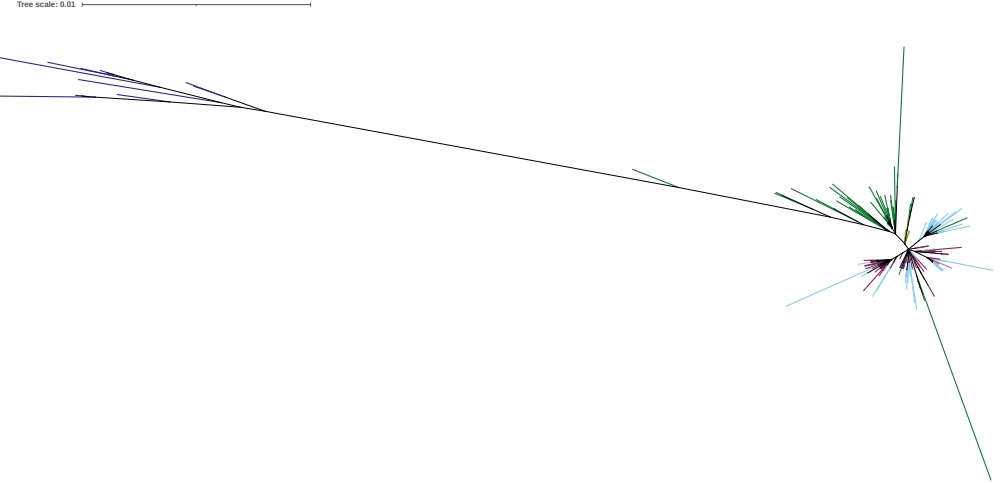

Tree scale: 0.01

**Figure 6 ZIKV phylogenetic tree produced with the best amino acid evolutionary model HIVb+F+R2 selected by IQ-TREE.** The branching lengths reflects the evolutionary distances between the proteomes. The geographic regions are indicated in colorblind-friendly hex code colors as #332288 (Africa), #117733 (Asia), #CC6677(Europe), #88CCEE (North America), #999933 (Oceania), and #882255 (South America).

**Table 1 Phylogenetic dependence statistic test results and its *p* value.**

| Axis ID | Abouheif's Cmean | p value | Pagel's Lambda | *p* value |
|---|---|---|---|---|
| PCA_dim1 | 0.8655691 | 0.001 | 1.010549 | 0.001 |
| PCA_dim2 | 0.9582174 | 0.001 | 1.003788 | 0.001 |
| t-SNE_dim1 | 0.9440515 | 0.001 | 1.010642 | 0.001 |
| t-SNE_dim2 | 0.9719502 | 0.001 | 1.003857 | 0.001 |
| UMAP_dim1 | 0.9792323 | 0.001 | 1.046470 | 0.001 |
| UMAP_dim2 | 0.9763210 | 0.001 | 1.110222 | 0.001 |
| Autoencoders_dim1 | 0.9050112 | 0.001 | 1.001254 | 0.001 |
| Autoencoders_dim2 | 0.9417386 | 0.001 | 1.022134 | 0.001 |

The phylogenetic clustering was related to the first two-dimension coordinates obtained by the four DRs. A phylogenetic correlogram analysis showed to some extent significant positive correlations of the coordinates with smaller branch lengths and significant negative correlations with larger branch length (Figs. 7A, 7B, 7C and 7D). Additionally, the Abouheif's Cmean and Pagel's Lambda tests (Table 1) showed also significant phylogenetic dependency with the dimension coordinates. These results confirm that UL retains the evolutionary findings as the phylogenetic gold standard methods without the need of an underlying evolutionary method.

## DISCUSSION

The application of clustering techniques as the phylogenetic trees is the gold standard in the field of sequence evolution studies. In this work, the utility of UL clustering algorithms in comparison with the phylogenetic method was explored. The UL approach applied herein

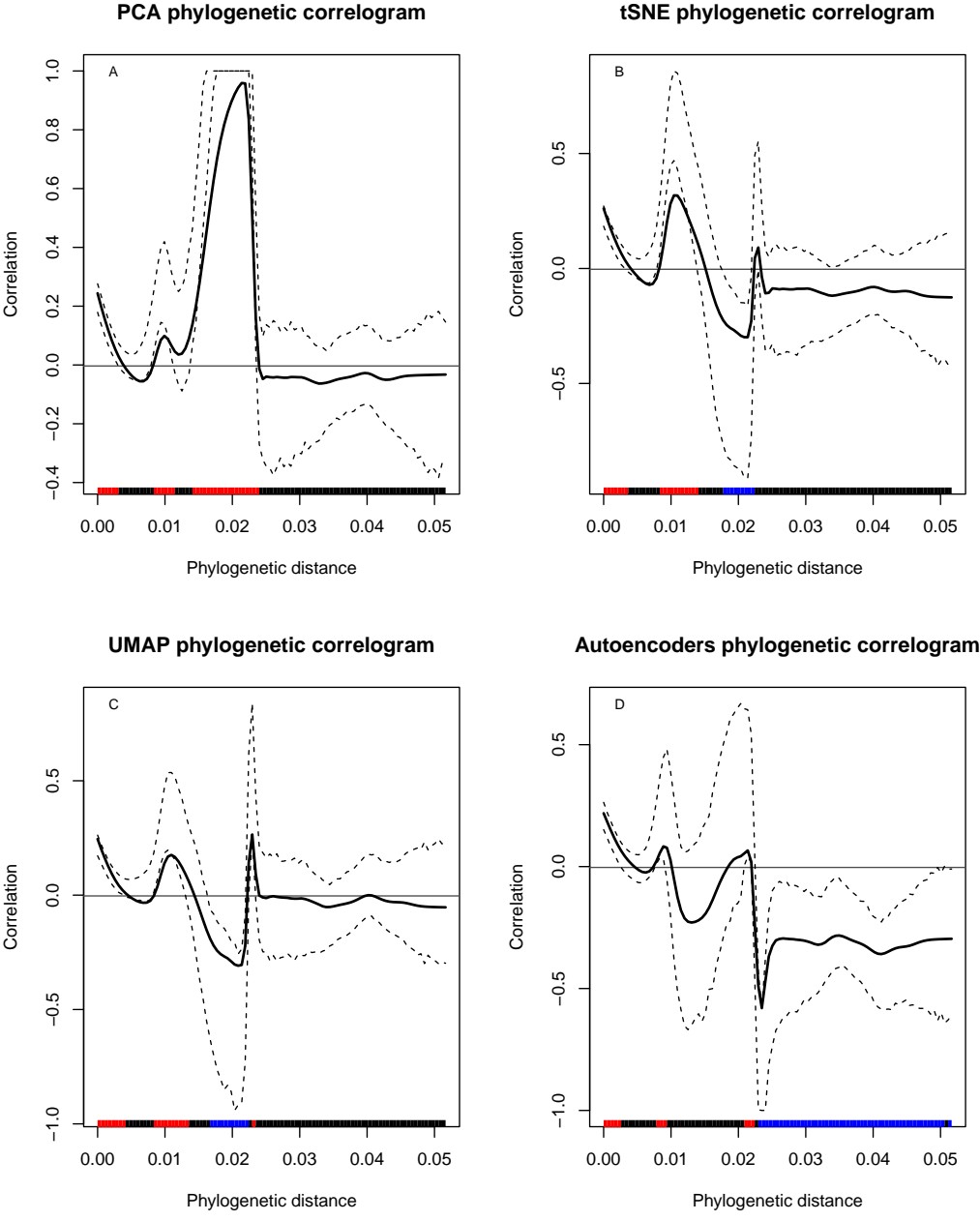

**Figure 7** **(A–D) Phylogenetic correlograms of DRs clustering data and the phylogenetic branch lengths.** Each panel showed the correlation values ($y$-axis) with its confidence interval (dashed lines) in relation with the branching length ($x$-axis). The bottom red bar indicates significant positive correlation meanwhile blue bar indicates negative significant correlation and the black bar means no significant correlations.

focuses on the amino acid polymorphic data extracted from the multi-alignments. Firstly, our URF model was optimized to the data, and the best hyperparameters were searched by five-fold cross-validation and tree repeats. This procedure is recommended to improve the parameters for the machine learning algorithm in the further prediction tasks (*Bradshaw*

*et al., 2023*). Thus, the parameter for the number of trees was settled in 1,500, above the 500 trees threshold previously suggested as enough to reduce the computational cost and maintain the best performance (*Oshiro, Perez & Baranauskas, 2012*). On the other hand, the maximum number of predictors to produce the best split in the random forest tree building process was below the square root of the total predictors, previously suggested to be sufficient to prevent biased predictions (*Afanador et al., 2016*). These parameters allow the application of the URF to produce the dissimilarity matrix in which each covariate (*i.e.,* the amino acid polymorphic sites) is weighted without the need of data normalization since it only depends on the feature ranks produced by the RF. Thus, the more correlated a covariate with the others, the more it will be considered in the forest dissimilarity matrix (*Shi et al., 2005*). In this way the dissimilarity matrix inherently allows the mapping of each proteome as related points in a two-dimensional space through the application of different DR algorithms.

In a related context, the application of DR algorithms requires two main critical steps to identify the meaningful hidden structures on the data, the internal and external validation. The internal validation through the elbow curve analysis using the silhouette coefficients as score showed consistency with those results of other studies, in which UMAP outperforms the other DRs (*Trozzi, Wang & Tao, 2021*; *Yang et al., 2021*). Thus, UMAP could be the first option to apply in this type of studies. Remarkably, the four applied DR approaches clearly showed similar host but particularly geographical clusters in the external validation (Fig. 4). Phylogenetic analysis suggested that ZIKV originates from Africa. The virus was circulating in Southeast Asia and the South Pacific regions since the 1960s, and it was hypothesized that there was acquired adaptive evolution that makes the virus more competent in urban cycle transmission in the vector *Aedes aegypti* (*Weaver et al., 2016*; *Hung & Huang, 2021*). From these regions, the virus spread to the other areas with temperate and tropical climates, such as the Americas. This evolutionary tracking description was also observed in the low dimensional clustering patterns found by the UL, even the imported viruses were visually discovered by the UL. All these findings were also detected in a DENV-2 large external dataset (Fig. 5), confirming the applicability of the unsupervised analysis in evolutionary bioinformatics studies.

Additionally, to provide an objective comparison of the UL clustering results with the phylogeny, a statistical correlation analysis was performed. The phylogenetic correlograms and the statistical test of phylogenetic dependence require external traits for analysis. To this end, both approaches were applied using the two-dimension coordinates as traits. The phylogenetic correlograms showed that the first two dimensions correlate with phylogenetic tree branches. In these analyses, the smaller branch lengths correlate with the tight cluster coordinates, which resembles the closest geographical terminal branches observed in the phylogeny. On the contrary, larger terminal branches showed significant negative correlations with larger cluster coordinates as observed for the dimension coordinates of the African clusters. Additionally, it tested the null hypothesis of no correlation for the two-axis coordinates with the phylogeny. The Abouheif's Cmean and Pagel's Lambda tests, which are the best statistical measures for this analysis, were used (*Münkemüller et*

*al., 2012*). Both statistics showed that the coordinates depended on the phylogenetic tree, confirming the successful application of UL.

**Limitations/validity:** The visualization and operation of large data sets are known to be difficult using phylogenetic clustering methods (*Menardo et al., 2018*). On the contrary, the UL techniques applied herein have the advantage of their use in both large and small data sets (*Fondrie & Noble, 2020*). Thus, these UL could be adequate to study these evolutionary issues, directing further in-depth manageable phylogenetic datasets. On the other hand, it is recognized the limitation of UL are the unpredictable results in the absence of data quality that could lead to inaccuracies in the clustering analysis interpretation (*Wang & Biljecki, 2022*). However, the availability of metadata information for geography further confirmed that our method works.

## CONCLUSIONS

In this work we demonstrated the utility of the UL approach to study evolutionary trends hidden in whole ZIKV proteome sequences. These UL techniques uncover host and particularly geographical clusters without the need of an underlying evolutionary model and were significantly comparable to the phylogenetic tree. Therefore, these UL algorithms could be another practical evolutionary analytical technique to trace the dispersal of viral pathogens. As the genomic and proteomic data increase, the evolutionary model-based study could become inefficient. In this scenario, the unsupervised analysis strategies would be useful. To manage the increasing number of large-scale data, advanced DR adapted for memory limitations and parallel computing needs to be developed. Moreover, the algorithm application needs to be user-friendly by developing better visualization-based exploratory tools that do not require programmatical expertise. Moreover, the inclusion of evolutionary data in the learning process must be also considered, as the incorporation of semi-supervised learning algorithms or deep learning algorithms has been recently reported. (*Voznica et al., 2022*; *Thompson et al., 2024*). Hence, curated evolutionary datasets are also needed to conform their use as benchmarking standards for algorithm performance evaluation. In this sense, the research area on unsupervised learning could help to study more complex evolutionary issues.

**Abbreviations**

| | |
|---|---|
| **DENV2** | dengue type 2 |
| **DRs** | Dimensional Reduction |
| **PCA** | Principal Component Analysis |
| **UMAP** | Uniform Manifold Approximation and Projection |
| **t-SNE** | t-distributed Stochastic Neighbor Embedding |
| **UL** | Unsupervised Learning |
| **URF** | Unsupervised Random Forest |
| **ZIKV** | zika virus |

### Funding

This work was supported by "Secretaria de Investigación y Posgrado del Instituto Politécnico Nacional" PRORED-2024. Edgar E. Lara-Ramírez holds a scholarship from the "Programa de Estímulos al Desempeño de los Investigadores" (EDI-IPN). The funders had no role in study design, data collection and analysis, decision to publish, or preparation of the manuscript.

### Grant Disclosures

The following grant information was disclosed by the authors:
Secretaria de Investigación y Posgrado del Instituto Politécnico Nacional: PRORED-2024.
Programa de Estímulos al Desempeño de los Investigadores: PRORED-2024.

### Competing Interests

The authors declare there are no competing interests.

### Author Contributions

- Edgar E. Lara-Ramírez conceived and designed the experiments, performed the experiments, analyzed the data, performed the computation work, prepared figures and/or tables, authored or reviewed drafts of the article, and approved the final draft.
- Gildardo Rivera analyzed the data, authored or reviewed drafts of the article, and approved the final draft.
- Amanda Alejandra Oliva-Hernández performed the experiments, authored or reviewed drafts of the article, and approved the final draft.
- Virgilio Bocanegra-Garcia performed the experiments, authored or reviewed drafts of the article, and approved the final draft.
- Jesús Adrián López analyzed the data, authored or reviewed drafts of the article, and approved the final draft.
- Xianwu Guo conceived and designed the experiments, performed the experiments, analyzed the data, prepared figures and/or tables, authored or reviewed drafts of the article, and approved the final draft.

### Data Availability

The R code and data are available in the Supplementary Files.

### Supplemental Information

Supplemental information for this article can be found online at http://dx.doi.org/10.7717/peerj-cs.2443#supplemental-information.

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
