# Peer review of "Unsupervised learning analysis on the proteomes of Zika virus"

_PeerJ Computer Science, doi:10.7717/peerj-cs.2443_

## Round 0.1 · original submission · Major Revisions

The reviewers have substantial concerns about this manuscript. The authors should provide point-to-point responses to address all the concerns and provide a revised manuscript with the revised parts being marked in different color.

Reviewer 1 ·

Basic reporting

N0.

Experimental design

This research paper uncover hidden patterns from polymorphic amino acid sites extracted on the proteome ZIKV multialignments, without the need of an underlying evolutionary model via four dimentional reduction models, and revealed speciûc host and geographical clustering patterns for ZIKA virus.
However, it would be better to include a workflow chart (as the first figure) for this research to enhance clarity and guide readers through the process.

Validity of the findings

1.In original dataset, is there any missingness? How do you handle the missingness if exists?
2.How do you ensure the uniquesness of the results in dementional reduction when using UMAP and t-SNE?
3. What is the best number of groups in k-means clustering process when evaluated withthe silhouette coefficient? Is it exactly 6 or not?

Additional comments

No.

Cite this review as

Reviewer 2 ·

Basic reporting

The paper titled "Unsupervised learning analysis on the proteomes of Zika virus" investigates the use of unsupervised learning (UL) algorithms to analyze the proteomes of the Zika virus. The author try to make a point that this study explores the potential of UL algorithms, which do not require labeled training data, to reveal additional insights, which differ itself with the traditional methods. Specially, the authors applied Unsupervised Random Forest (URF) along with dimensional reduction algorithms such as Principal Component Analysis (PCA), Uniform Manifold Approximation and Projection (UMAP), t-distributed Stochastic Neighbor Embedding (t-SNE), and AutoEncoders (AE) to analyze polymorphic amino acid sites from ZIKV proteome alignments. The study found that the four UL algorithms could identify specific host and geographical clustering patterns for ZIKV. They were also capable of detecting imported viruses within these geographical clusters. Among the dimensional reduction techniques, UMAP performed the best. The UL axis coordinates showed significant correlation with phylogenetic tree branch lengths and demonstrated significant phylogenetic dependence in the Abouheif Cmean and Pagel’s Lambda tests (p < 0.01), indicating that these UL methods offer comparable performance to traditional phylogenetic methods.
Comments
I think the paper is generally well-structured, but I believe polishing the writing and providing better explanations will benefit the paper more. I enjoyed reading it and suggest a minor revision.
1. Motivation:
o My understanding is that motivation is still part of the introduction. So, I suggest mentioning more previous research using unsupervised learning for more specific biological data, such as genomes and proteomes.
2. Methods:
o Model Section: Your description of leveraging URF models for data analysis could benefit from a pseudo-code section. The current format is confusing and hard to follow.
o What’s the rationale for choosing URF as the primary method, given there are many other methods available nowadays? How did you decide on the optimal hyperparameters for URF? Are your clustering results still robust when using a different set of hyperparameters?
3. For all the major R packages that you used, please cite them properly.
4. Avoid using abbreviations like AE; instead, use the full term "Autoencoder" for clarity.
5. I understand that Figure 1 is for visualization and Figure 2 supports Figure 1 as metrics for measuring clustering separation. However, an additional analysis measuring the similarity and diversity between different regions would be beneficial. You cannot get that kind of information just from UMAP, etc. I suggest calculating the Euclidean distance between the centroids of individual groups and providing a similarity matrix.
6. The authors also need to provide some evaluation or summary on how well their URF model fits the data.

Experimental design

no comment

Validity of the findings

The authors also need to provide some evaluation or summary of how well their URF model fits the data.

If you want to claim that UL algorithms could be practical evolutionary analytical techniques to track the dispersal of viral pathogens, experiments on an external dataset are encouraged.

Additional comments

The other thing I want to mention is that the authors might want to tone down the claims in the manuscript a little bit. I don't fully agree with the level of novelty that the authors claim.

Cite this review as

Reviewer 3 ·

Basic reporting

Here are some specific comments for the manuscript:
1. A list of all abbreviations is recommended. More details in the caption are needed, e.g., the explanation of x and y axes in Figure 1.
2. What is the rationale for using the proteome for clustering?
3. Please also discuss the traditional approaches used for clustering as well as the pros and cons to show the necessary requirements for the development of a new clustering tool.
4. Please include the proteome data used in the manuscript.

Experimental design

Overall, the authors clearly described the four algorithms used in the manuscript with sufficient details, and here are some specific comments:
1. Please provide an overall workflow for the clustering.
2. How would the new tool be compared to a traditional clustering tool? Are there any additional significant insights that could be obtained using the presented method? What are the pros and cons? How long does it take for the clustering process?

Validity of the findings

The results and conclusions are well stated and supported by the presented data. However, how would the authors comment on the future directions and how about the application of the presented algorithms in other fields?

Additional comments

The authors conducted four unsupervised learning algorithms on the ZIKV proteome obtained from the virus variation database. They successfully identified specific host and geographical clustering patterns, showing results comparable to the phylogenetic tree. The work presented an alternative practical evolutionary analytical technique to trace the dispersal of viral pathogens. Here are some specific comments:

Cite this review as

---

## Round 0.2 · accepted · Accept

Reviewers are satisfied with the revisions, and I concur to recommend accepting this manuscript.

Reviewer 1 ·

Basic reporting

I am satisfied with the improvement the authors made in current version of paper draft.

Experimental design

No.

Validity of the findings

No.

Additional comments

No.

Cite this review as

Reviewer 2 ·

Basic reporting

The authors have addressed all the requested revisions thoroughly.
1. They have provided examples of unsupervised learning in biological data, improved their method description, added pseudocode and workflow charts, cited R packages correctly, replaced abbreviations with full terms, and performed additional analyses, including a similarity matrix and evaluation of clustering robustness.
2. Additionally, they have toned down the claims of novelty as suggested.

Based on these comprehensive updates,I recommend accepting the manuscript.

Experimental design

NA

Validity of the findings

NA

Additional comments

NA

Cite this review as

Reviewer 3 ·

Basic reporting

All concerns have been addressed, and the paper is now well-structured and clearly articulated following the revision.

Experimental design

All concerns have been addressed, and the paper provides a clear and detailed description of the four algorithms used in the manuscript. Additionally, the new functions to monitor time consumption in each analytical section are beneficial.

Validity of the findings

All concerns have been addressed.

Additional comments

N/A

Cite this review as